# Helheim Glacier ice velocity variability responds to runoff and terminus position change at different timescales

Lizz Ultee [1,2,3] ✉, Denis Felikson [4,5], Brent Minchew[3], Leigh A. Stearns [6] & Bryan Riel[3,7]

The Greenland Ice Sheet discharges ice to the ocean through hundreds of outlet glaciers. Recent acceleration of Greenland outlet glaciers has been linked to both oceanic and atmospheric drivers. Here, we leverage temporally dense observations, regional climate model output, and newly developed time series analysis tools to assess the most important forcings causing ice flow variability at one of the largest Greenland outlet glaciers, Helheim Glacier, from 2009 to 2017. We find that ice speed correlates most strongly with catchment-integrated runoff at seasonal to interannual scales, while multi-annual flow variability correlates most strongly with multi-annual terminus variability. The disparate time scales and the influence of subglacial topography on Helheim Glacier's dynamics highlight different regimes that can inform modeling and forecasting of its future. Notably, our results suggest that the recent terminus history observed at Helheim is a response to, rather than the cause of, upstream changes.

In recent decades, several glaciers draining the Greenland Ice Sheet have accelerated, increasing their contribution to global mean sealevel rise[1–3]. The observed acceleration of outlet glaciers and the ice sheet interior has been attributed to warmer ocean waters melting glacier fronts[4,5] as well as increased surface melt[6,7]. Numerical models and indirect observations indicate that increasing runoff could enhance solid ice loss by lubricating the glacier bed and warming the ice such that it deforms more readily[8–12]. However, in situ observations of the Greenland Ice Sheet margin have found limited evidence for annual-scale acceleration of ice flow driven by increasing runoff[13,14]. At marine outlets including Helheim Glacier, observations show that ice flow speed (and therefore mass discharge) correlates most strongly with iceberg calving activity rather than runoff[6,15–18].

Helheim Glacier is one of the highest-flux outlets of the Greenland Ice Sheet, in recent years matching or surpassing Sermeq Kujalleq (Jakobshavn Isbræ) in solid ice discharge[19]. Its dynamics through the early 21st century showed pronounced variability, including episodes of multi-annual retreat and readvance[3,15,20] and net mass gain while most Greenland outlet glaciers were losing mass[21]. Sediment records from the past century suggest that Helheim responds to atmospheric and oceanic variability on time scales of a few years[22], highlighting the importance of understanding its dynamics on seasonal to multi-annual time scales. The high ice flux through Helheim Glacier[23,24], its recent variability[15,20,25], and its sensitivity to short-term variation in climate forcings[22,26] motivate a quantitative comparison of hypothesized controls on velocity variability.

Processes contributing to velocity variability operate at different time scales. For example, fracture-driven changes in stress balance can be nearly instantaneous and propagate rapidly, shaping velocity on the order of hours to days[16,27,28], while changes in the subglacial drainage system may take days to months[29–33] and response to changing upstream snow accumulation can take many years[34–36]. Observations

[1]Dept. of Earth & Climate Sciences, Middlebury College, Middlebury, VT, USA. [2]School of Earth & Atmospheric Sciences, Georgia Institute of Technology, Atlanta, GA, USA. [3]Dept. of Earth, Atmospheric, and Planetary Sciences, Massachusetts Institute of Technology, Cambridge, MA, USA. [4]Cryospheric Sciences Laboratory, NASA Goddard Space Flight Center, Greenbelt, MD, USA. [5]Goddard Earth Sciences Technology and Research II, Morgan State University, Baltimore, MD, USA. [6]Department of Geology, University of Kansas, Lawrence, KS, USA. [7]School of Earth Sciences, Zhejiang University, 310027 Hangzhou, China. ✉e-mail: eultee@middlebury.edu

that permit a detailed understanding of one process – such as intensive field study of a calving front – may not be sufficient to contextualize influences from processes operating at other scales. Accounting for the relative influence of each process, for example to develop accurate predictive models, requires synthesizing observations and inference across time scales. Here, we apply the flexible time series analysis tools developed by Riel et al.[37] to publicly available velocity fields[38] and correlate the results with temporally dense climate model output[39,40] and terminus observations[41] to study the forcings of and responses to velocity variability at Helheim over multiple temporal scales Fig. 1.

## Results

### Seasonal to interannual velocity variability responds most strongly to runoff

The normalized, single-differenced cross-correlations with ice surface speed are distinct for each variable. The weakest cross-correlations, in terms of mean magnitude of maximal values along the flowline, are with catchment-integrated surface mass balance (Fig. 2, left column).

For that variable, correlations with velocity range from −0.17 near the terminus to 0.13 farther up the flowline. Cross-correlation of ice surface speed with catchment-integrated runoff (Fig. 2, center column) is stronger, ranging from −0.20 to 0.25. Terminus position (Fig 2, right column) also shows comparatively strong cross-correlations with velocity. The strongest correlation is −0.22, found near the terminus, and the strongest positive correlation is 0.18, found 14 km upstream from the terminus. However, the strongest cross-correlations with terminus position are found at 0 lag at all points on the lower 10 km of the glacier trunk. This suggests that terminus position and velocity change simultaneously, or influence each other over time scales shorter than the temporal resolution of our data, such that a clear forcing on velocity by terminus position is not apparent at this scale.

At every point, the magnitude of strongest cross-correlation with velocity is larger for runoff than for surface mass balance, on average 1.2 times larger over the flowline, and this difference exceeds the significance limit for most points on the flowline. The cross-correlation between terminus position and velocity is similar in magnitude to that

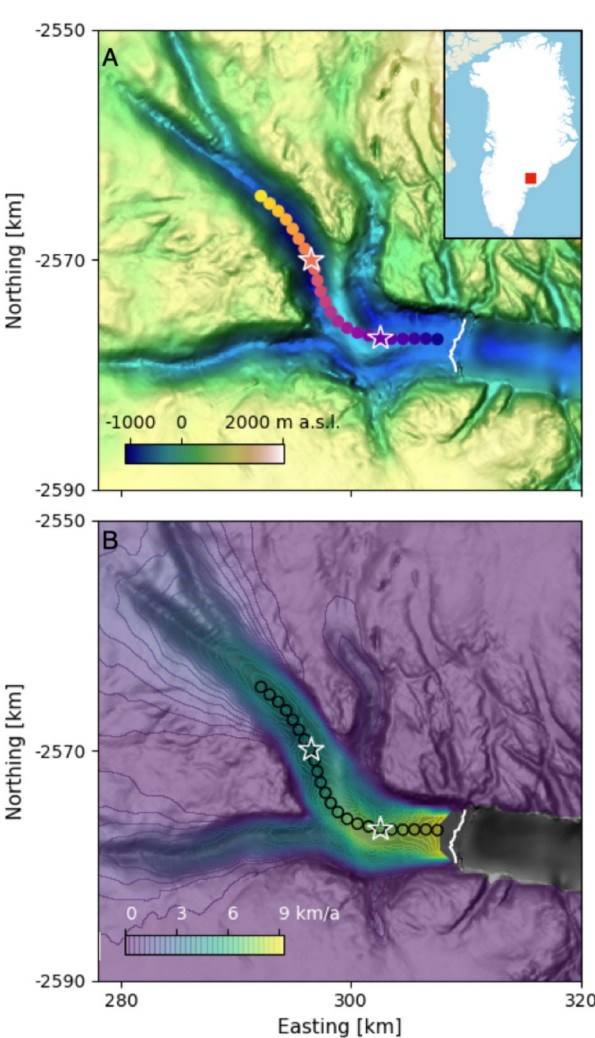

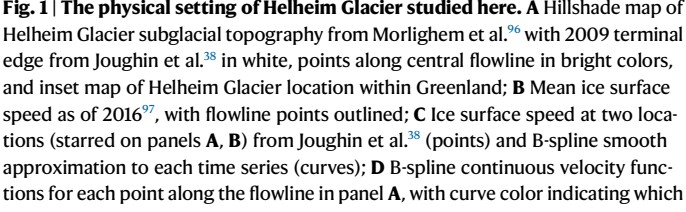

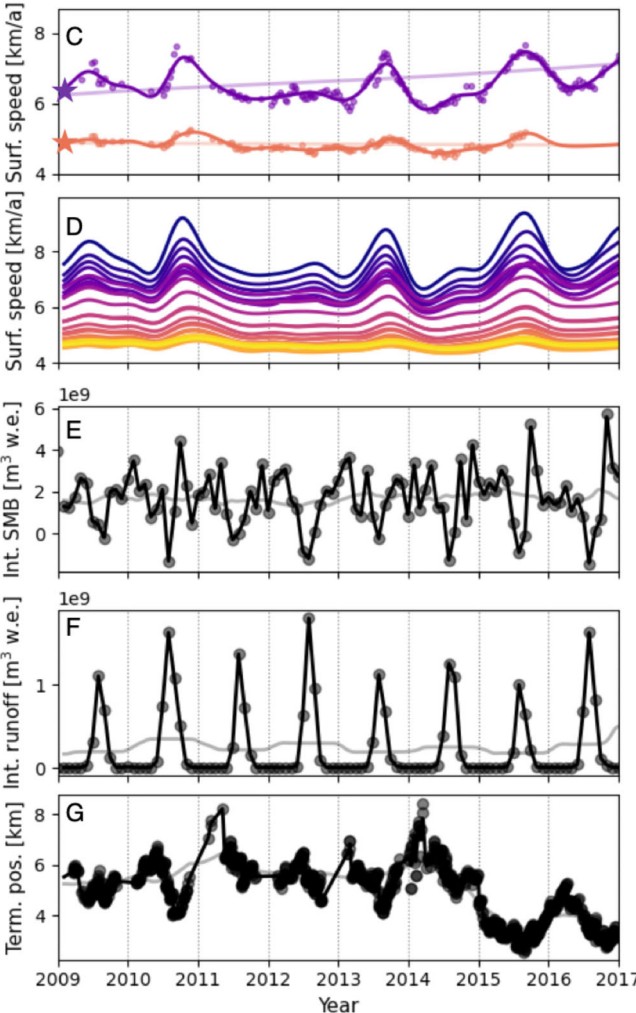

**Fig. 1 | The physical setting of Helheim Glacier studied here. A** Hillshade map of Helheim Glacier subglacial topography from Morlighem et al.[96] with 2009 terminal edge from Joughin et al.[38] in white, points along central flowline in bright colors, and inset map of Helheim Glacier location within Greenland; **B** Mean ice surface speed as of 2016[97], with flowline points outlined; **C** Ice surface speed at two locations (starred on panels **A**, **B**) from Joughin et al.[38] (points) and B-spline smooth approximation to each time series (curves); **D** B-spline continuous velocity functions for each point along the flowline in panel **A**, with curve color indicating which point is represented; **E** Catchment-integrated surface mass balance from RACMO; **F** Catchment-integrated runoff from RACMO[40]; and **G** Width-averaged terminus position, relative to a fixed gate on the glacier (larger numbers indicate advance). In panels **E**–**G**, data from the original source is plotted as points, and dark lines show the values of 1d-interpolated functions used to determine signal cross-correlation. In panels **C** and **E**–**G**, light curves show the long-term-varying component of each signal. Long-term-varying velocity is shown with a zoomed y axis in Fig. S5.

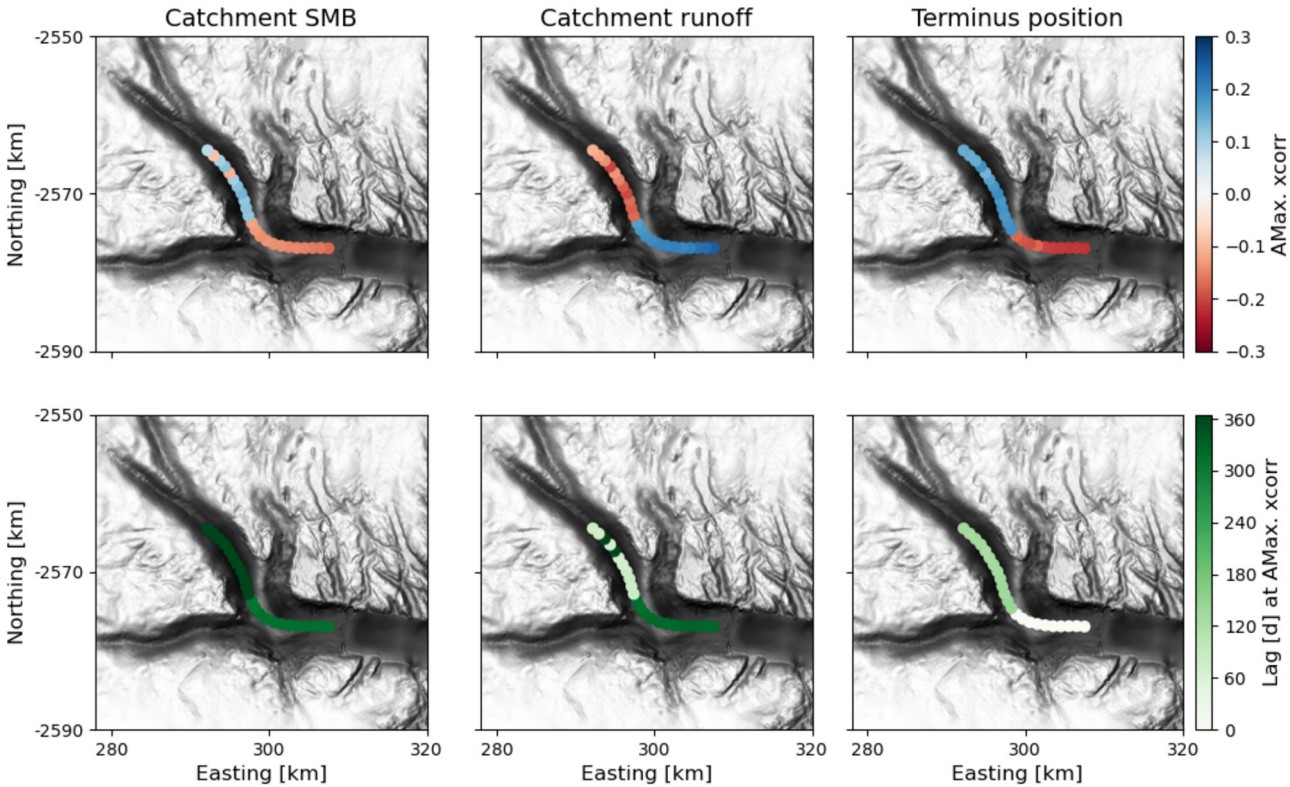

**Fig. 2 | The cross-correlation of largest absolute value ("AMax. xcorr") (top row) between ice surface speed and each variable (columns), and the lag in days (bottom row) at which that cross-correlation is found.** Circles indicate values that are significant at the 95% confidence level; all values plotted here are significant, in contrast with Fig. S6. For along-flow view, see Fig. S7.

of runoff, but the former may instead be a response to velocity changes. We infer that runoff is at least as important as terminus position in controlling seasonal to interannual ice surface velocity variability along the main trunk of Helheim Glacier.

### No year in which terminus position is more important than runoff

Because Helheim Glacier is a complex system that changes over time, the multi-year bulk analysis of the preceding section may not capture important interannual changes in the dominant sources of its velocity variability. To study year-to-year changes in more detail, we computed the cross-correlation between single-year subsets of the variables we studied above. Cross-correlations of these single-year subsets are generally stronger than those found over the full time period.

The patterns of cross-correlation between single-year sections of the signals vary from year to year, as shown in Fig. 3. For example, in 2009 the cross-correlation between runoff and surface speed for all points along the flowline is strongest at a lag of around 60 days, with a significant minimum following at longer lag times. In 2010, 2011, 2014, and 2016, an initial negative correlation around 60 days lag is followed by a small maximum around 200 days. In 2012, cross-correlations with velocity at different points along the flowline show different patterns of maxima and minima, for both surface mass balance and runoff. There are statistically significant correlations between runoff and surface speed every year, for every point along the flowline.

We find that the normalized cross-correlation of terminus position and ice surface speed is low at every point along the flowline and for almost every year 2009–2016. Only 4 of the 8 years we study show correlations significantly different from zero for one or more points along the flowline. Stronger correlations with terminus position, some of which are significant, are evident for

negative lags (Fig. S3), indicating that terminus position may be responding to velocity variation rather than vice versa. For every year and every point we study, the positive-lag correlation of ice surface speed with catchment-integrated runoff is stronger than that with terminus position. In most years and for most points, the correlation with runoff is stronger than that with terminus position for both positive and negative lags (Fig. S3).

### Multi-annual velocity variability correlates with terminus position

We see a strong and statistically significant correlation between the long-term-varying components of ice speed and terminus position. The correlation between these two component signals is much stronger than between the corresponding full signals (Figs. 2, 4 and S6), with values along the lower trunk averaging −0.8, all for non-negative lags. A cross-correlation stronger than that for the full signals is also seen for long-term-varying surface mass balance, ranging from −0.54 to 0.54, but due to strong autocorrelation none of the values is significantly different from 0. The correlation between long-term-varying components of ice speed and runoff is comparable to that between the full signals, ranging from −0.29 to 0.29, but not significantly different from 0. We infer that terminus position variability is the only one of our variables that is important for Helheim Glacier's dynamics at multi-annual time scales (here 2009–2017).

### Subglacial topography modulates velocity response to each variable

The flowline we examine flows through a trough with a pronounced ridge in its subglacial topography. The ridge creates a steep along-flow thickness gradient as well as a lateral constriction (Fig. 4B). For all three variables, the flowline separates into two segments with opposite sign of maximum cross-correlation. We find changes in

sign of absolute maximum cross-correlation with velocity at 14 km upstream from the terminus – coincident with the upstream edge of the subglacial ridge (Figs. 2, 4 and S6). We also find step changes in

the lag at peak cross-correlation aligned with the ridge. The spatial pattern of cross-correlation is similar for both seasonal and multi-annual signals (earlier Results sections). These patterns suggest that

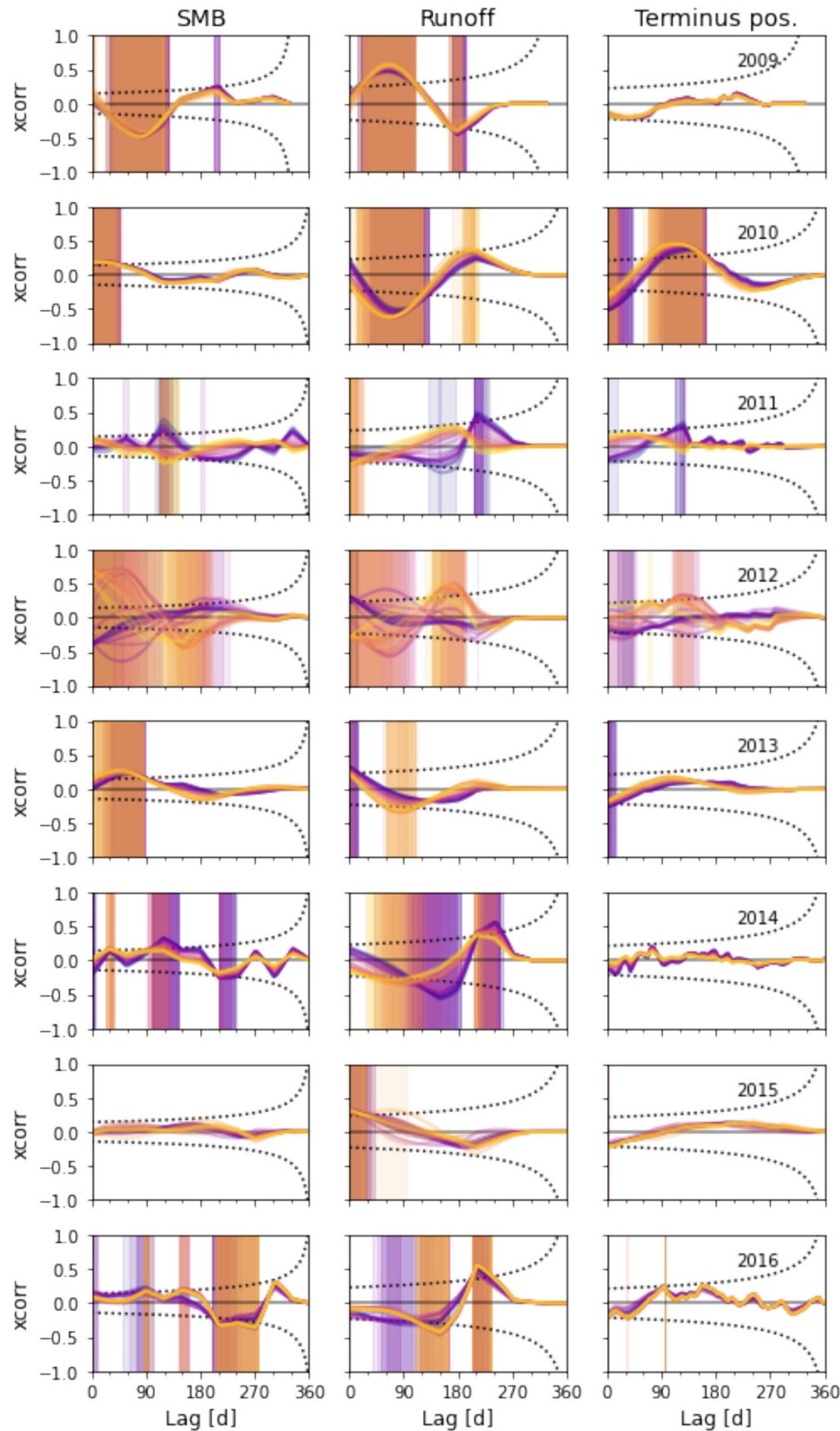

**Fig. 3 | Annual patterns of cross-correlation between surface speed and system variables for (left) surface mass balance, (center) runoff, and (right) terminus position, sampled at 1 km intervals along the flowline shown in Fig. 1.** Dotted curves indicate 95% confidence intervals around XCorr($f$, $v$) = 0, modified for autocorrelated data as described in Methods section; shading indicates statistically significant difference from zero. Color of lines and shading indicates location of the example point along the flowline, matching Fig. 1A, C, and D.

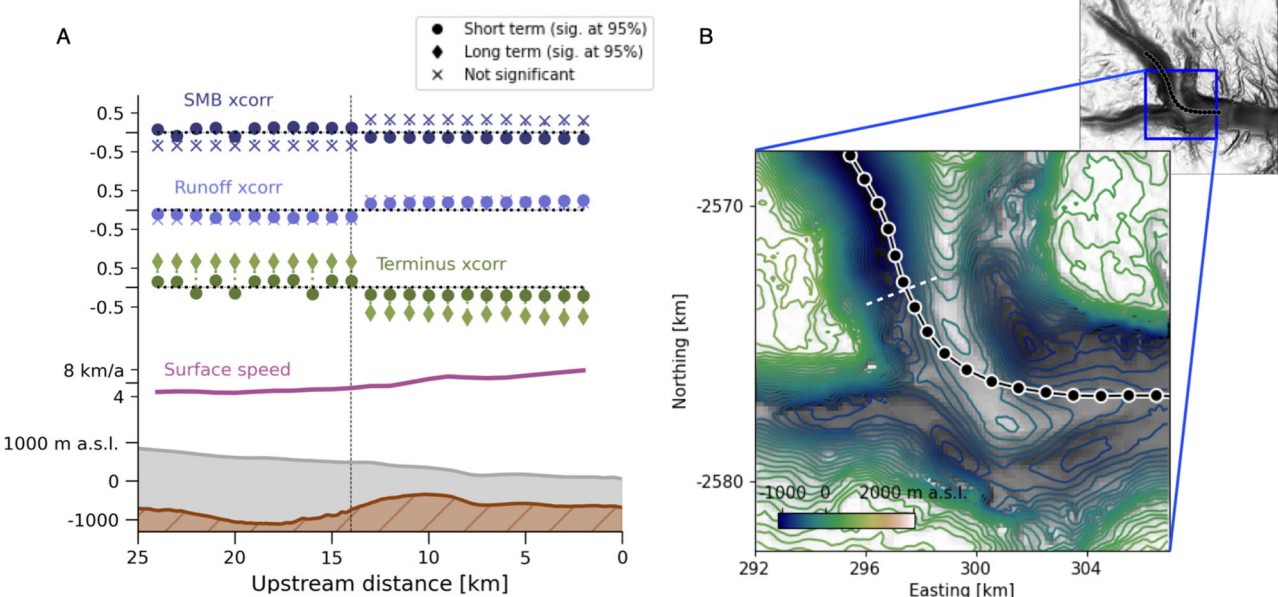

**Fig. 4 | Influence of a subglacial ridge on Helheim Glacier dynamics. A** Ice speed cross-correlation with each variable tested, for each point along the flowline, vertically offset for legibility. Variable labels coincide with zero cross-correlation and minor ticks indicate XCorr($f$, $v$) = ± 0.5. Darker circles are cross-correlations of the full signals (as reported in Fig. 2 and the first Results section). Lighter diamonds show results filtered to isolate long-term variability (as in Results section header "Multi-annual..." and Fig. S6). Results not significantly different from 0 are assigned a cross marker, as in Fig. S6. Lower portion shows bed topography (brown), ice surface (gray), and mean surface speed (purple) along the flowline. Vertical marker indicates position of sign changes in cross-correlation for multiple variables.
**B** Enlarged contour map of the Helheim Glacier trough around the bedrock bump. Outlined points show locations where velocity was extracted along the flowline; a dashed white line across the direction of flow indicates the approximate location of the dashed line in panel **A**. Background image is a black and white hillshade of the topography as in Fig. 2; contours show intervals of approximately 60 meters elevation. Contour colormap and flowline points (black) are consistent with Fig. 1A.

the dynamics of the upstream and downstream segments of the flowline are fundamentally different from one another. We interpret that the bedrock ridge is an obstacle to the propagation of traveling waves[35,42,43]. For example, adjustment in the glacier stress balance due to changes in ice accumulation (related to surface mass balance) would propagate as a kinematic wave from the accumulation zone to the ablation zone, and that wave could be obstructed by the vertical and lateral constriction of the ridge. Similarly, changes at the terminus can initiate upstream-propagating kinematic[44] or dynamic waves[45], which could be slowed by the steeper bed slopes around the ridge. We would also expect wave-like propagation of changing basal friction due to seasonal runoff input. The bedrock ridge modifies bed slope and ice overburden pressure, which will modify the hydraulic potential gradient and therefore also the direction of subglacial water flow around it.

## Discussion

Our analysis illustrates that Helheim Glacier is a dynamic system with more than one important control on its velocity. We find that seasonal-scale variations in ice surface speed respond more strongly – that is, have larger cross-correlation values at strictly positive lag times – to catchment-integrated runoff than to terminus position, for the full period 2009-2017 (Fig. 2) and for every year in it (Fig. 3). At the multi-annual scale we find stronger correlation with terminus position than with runoff (Fig. 4), in agreement with earlier work relating ice velocity to ice thickness and glacier terminus position on Alaskan tidewater glaciers[46,47]. Our results support previous findings that increasing meltwater supply can enhance seasonal speedups in ice flow, but does not contribute to multi-annual acceleration (summarized in ref. 14). Our analysis also supports the hypothesis of Enderlin et al.[48] that distinct variables could drive different timescales of velocity variability at Columbia Glacier, Alaska.

It is tempting to attribute physical meaning to the lag times of strongest cross-correlation. For example, the 60-day lag time in cross-correlation between runoff and ice surface speed that we observe in many years (Fig. 3) could reflect the time required for water input to induce large-scale changes in subglacial water pressure, and therefore ice sliding velocity. That interpretation would agree with the model results of Poinar et al.[49], who applied low-elevation meltwater input to an idealized Southeast Greenland outlet glacier and found that domain-averaged subglacial water pressure peaked 60 days into the melt season. The range of lag times for the handful of significant cross-correlations between speed and terminus position, 3–100 days, also aligns with theoretical results on the speed of dynamic wave propagation upstream from a calving terminus[45]. However, we caution that single-differencing the signals in Results "Seasonal to interannual..." - "No year..." produces different phase shifts in signals with different shapes (Supplementary Note S3 and Fig. S4), which complicates the interpretation of lag times. We therefore refrain from more detailed interpretation of the lag times.

The correlation of runoff with seasonal-scale velocity variation described in Results "Seasonal to interannual..." is consistent with observations of land-terminating margins of the Greenland Ice Sheet[6] and some marine outlets on Greenland's west coast[50,51], as well as inference of surface-melt-induced acceleration[52] and dynamic thinning[53] at Helheim. Because we analyze remote sensing and climate model output data, the significant cross-correlations we find are at seasonal and longer timescales; this complements the shorter-timescale correlations previously found in field data[16,52,54–56]. In agreement with Kehrl et al.[17], we find that 2010 and 2013 are the years for which ice surface speed at Helheim is most correlated with terminus position (Figs. 3 and S3). However, our quantitative comparison shows that, in all years, Helheim's speed is at least as correlated with runoff as with terminus position.

Our conclusions differ from previous studies including Moon et al.[57] and Vijay et al.[18], which infer that terminus changes are the strongest control on Helheim's velocity during most years. Moon et al.[57] studied Helheim terminus changes for the time period from 2009 to 2013 and Vijay et al.[18] studied Helheim terminus changes for the time period from 2015 to 2017. Despite contrasting conclusions, we identify some overlap in our findings. For example, Moon et al.[57], in their supplemental figure S35 find that terminus position was the primary control on velocity in 2010, and our results show that correlation between velocity and terminus position was strongest in 2010 (Fig. 3). In 2015 and 2016, we find strong correlation between velocity and runoff and weaker correlation between velocity and terminus positions, and in those years the time series presented in Vijay et al.[18], in their supplemental figure S28 show seasonal acceleration at Helheim appearing to coincide with melt onset and little change in terminus. We attribute our differing conclusions in part to contrasting methods: our approach focuses on statistically significant cross-correlations at a single glacier, while Moon et al.[57] and Vijay et al.[18] manually identified relationships within the context of a larger set of Greenland outlet glaciers.

The relative importance of each driver at Helheim Glacier likely does not translate to other outlets or other time periods. For example, our findings at Helheim contrast those of King et al.[58], who found that regionally aggregated trends in Greenland Ice Sheet discharge correlated most strongly to glacier front position. Ice velocity at Helheim may be unusually sensitive to catchment-integrated runoff because of the presence of a large firn aquifer that allows hydrofracturing of deep crevasses and enhances deformational ice motion[12,59]. The lower trunk of the glacier was also near flotation during the time period we study here[17], which could render it especially sensitive to both changing basal water pressure (runoff) and calving activity[28,55]. Finally, the spatial pattern of our results highlights the role of unique subglacial topography in shaping the dynamic response to forcing[44,48,60-63].

One explanation for the comparatively weak positive-lag correlation between seasonally varying ice surface speed and terminus position throughout our study period is that the sensitivity of surface speed to terminus position is itself determined by the terminus position[28], and that the terminus did not reach a hypothetical critical position during the time we observed. From 2009 to 2014, the observed terminus positions oscillated around a steady mean position at ~6 km forward of our reference position; a period of multi-annual retreat beginning in late 2014 reflects a multi-annual acceleration on the lower glacier trunk beginning around the same time (Fig. 1D, G). If the terminus had reached a critical position that increased the sensitivity of surface speed to terminus change, we would expect to see change in the correlation between those variables as terminus position changed over time. Instead we find that the annual cross-correlation between surface speed and terminus position is no stronger in 2015 and 2016 than in previous years (Fig. 3). In several years throughout the 2009–2017 period, there are significant cross-correlations between ice surface speed and terminus position, but they occur at negative lag times (Fig. S3). That suggests that terminus position is responding to, rather than driving, seasonal velocity variability.

A second explanation for the weak correlation between ice surface speed and terminus position is that iceberg calving is episodic and discontinuous. Field observations of Helheim Glacier at finer temporal scales than we study here have found that calving activity was an important control on velocity at the timescale of minutes to hours[16,54], and that runoff during the melt season contributes to daily velocity increases[52,55,56]. Thus, even with our temporally dense records – average 3 days between measurements – we may more realistically expect to see responses to runoff than to iceberg calving. Further, we analyze a width-averaged terminus position, which will not capture differing dynamic responses to iceberg calving at different points along the face. Extending our methodology to analyze the fine spatial and

temporal scales captured in field observations could provide a fuller picture of the forcings driving velocity variability (building on ref. 64, for example).

In this work, we have assumed that terminus position evolves independently from catchment-integrated runoff. This choice ignores the established connection between calving rate and subglacial discharge at the terminus[65-69]. Modeling efforts suggest that the calving response to subglacial discharge depends on the subglacial hydrologic system near the terminus, in particular whether melt is localized to channels[67,69,70]. Subglacial discharge also affects calving through its influence on the vertical pattern of submarine melt[67,71-75]. However, recent observations have found no evidence for a melt-induced enhancement of calving at Helheim Glacier, perhaps because of its broad and deep terminus[76]. As additional observations of the near-terminus environment become available, future work may apply multivariate statistical methods to assess whether runoff and calving activity reinforce or oppose one another in forcing ice surface velocity variability.

Although we have focused here on cross-correlations unique to Helheim during the 2009-2017 period, our methods can be used to investigate any glacier with a sufficient observational record. The statistical inference approach can also be applied to time series generated by different interpolation methods (e.g., ref. 77). We do not anticipate a strong dependence of the cross-correlation results on interpolation method (Supplementary Note S3). Nevertheless, future studies could benefit from more sophisticated data processing or more detailed numerical modeling than we have presented here. For example, we produced time series of both surface mass balance and runoff integrated over the whole Helheim catchment. We suspect that integrating both fields over only the portion of the catchment that is upstream of each point along the flowline would provide more spatially refined information. The advantages to be gained by additional data processing, however, should be weighed against the major uncertainties remaining from unconstrained processes. For example, our work has not accounted for the changing state of the en/subglacial hydrologic system over the melt season, nor for the effect of the Helheim catchment's firn aquifer[59] on both glacial hydrology and dynamics. To do so would likely require the use of a glacial hydrology model (such as ref. 78). Future efforts could also explore the use of Bayesian methods to account for uncertainties in climate-model-derived runoff, which can be as large as 20%[79,80].

The eight-year period of overlapping observations we studied, 2009–2017, necessarily limits our ability to resolve longer-term variability that may be important for glacier dynamics. For example, we found that surface mass balance had the lowest cross-correlations with velocity among the three variables we tested, but that finding does not preclude surface mass balance driving velocity variation at decadal and longer time scales. Ice core and radar reconstructions have shown that Greenland surface mass balance varies at multi-annual to multi-decadal time scales, correlated with the North Atlantic Oscillation, Atlantic Multidecadal Oscillation, and Greenland Blocking Index[81,82]. The known long-term variability of surface mass balance, combined with Helheim Glacier's large accumulation area and the long climatic response time of glacier flow[35,83], suggest that correlations between surface mass balance and ice velocity may well be found in records longer than those we have studied here.

Our results show that numerical ice flow modeling experiments will require multiple forcing mechanisms to capture the dynamics of Helheim Glacier. Several state-of-the-art studies, including the standard experiments performed by several numerical models as part of the the Ice Sheet Modeling Intercomparison for the Coupled Model Intercomparison Project Phase 6 ("ISMIP6", ref. 84), have used projections of outlet glacier terminus positions to force Greenland Ice Sheet mass change simulations[85,86]. Our results show that this approach is a good strategy for projections of multi-annual changes of

glaciers like Helheim. However, if future ice sheet modeling efforts seek to reproduce seasonal velocity changes, runoff forcing must be included. The continued development of subglacial hydrology models[33,78] and efforts to couple them with ice dynamics models[87,88] are therefore vital to refining our understanding of the future evolution of the Greenland Ice Sheet.

We have computed normalized cross-correlations between three catchment variables (surface mass balance, runoff, and terminus positions) and ice surface speed of Helheim Glacier, revealing the dominant controls on velocity variability at multiple time scales. We find that ice speed responds most strongly to catchment-integrated runoff at seasonal scale. The strongest cross-correlations between ice speed and terminus position occur at 0-day or negative lag times, suggesting that terminus position is responding to rather than driving seasonal velocity variability. At multi-annual scale, ice speed variability shows stronger correlation with terminus position change. We find distinct patterns in correlation along upstream and downstream portions of the glacier trunk, separated by a subglacial ridge. The time scale separation of major sources of variability, and the role of underlying topography, are important considerations in designing numerical ice flow simulators to project the future evolution of large outlet glaciers.

## Methods

### Inference framework
We investigate correlations between surface velocity and several factors hypothesized to drive its variability at seasonal to multi-annual scales. Limited time-dependent data precludes us from studying the effect of ice mélange, ocean temperature, and surface damage directly. Here, we assume that the primary effect of those three variables is on the rate of calving, and we restrict analysis of ocean-driven processes in the present study to the relationship between glacier terminus position and surface ice velocity. We focus our analysis on time scales of months to years. As such, we do not consider the flow response to individual calving events[89] or tidal variation[54,90], which have been described elsewhere. We also disregard any connection between terminus position and topography (via ice thickness), which has been explored in Kehrl et al.[17].

We investigate three factors varying in time (surface mass balance, runoff, and width-averaged terminus position) and one varying in space (subglacial topography). We examine subglacial topography qualitatively, rather than constructing a time series representation such as grounding-line depth, to allow a more holistic consideration of the glacier geometry beyond the near-terminus region. To quantify the strength of the temporal variables' relationship with velocity, we compute their cross-correlation as described below. We interpret the qualitative effect of local topography on velocity variation by analyzing spatial patterns in the cross-correlations computed for the temporal variables.

### Catchment data
We produce a one-dimensional time series for each catchment variable. We integrate monthly surface mass balance and runoff derived from Noël et al.[40] over the Helheim Glacier catchment defined by Mankoff et al.[91]. The time series of calving front position is a width-averaged distance from an upstream flux gate, identified from satellite imagery with variable temporal resolution[41]. For the present study of seasonal to multi-annual time scales, we apply a 10-day smoothing window to the terminus record. We trim all time series to the period for which data is available for all variables: 2009–2017. We interpolate a piecewise linear time-continuous function for each time series using the Interp1d class of SciPy v1.4.1 (ref. 92, and see Supplementary Note S4). Finally, we sample the interpolated function at a frequency matching the average time between velocity observations: approximately 3 days.

To isolate multi-annual variability from shorter-term signals (Results, "Multi-annual...") we apply a 1-year moving average filter to the surface mass balance, runoff, and terminus position data. The isolated long-term-varying components are shown as light curves in Fig. 1E–G.

### Producing temporally continuous velocity functions
We use frequent observations and spline interpolation to produce time-continuous estimates of ice surface velocity. We stack all available InSAR-derived glacier site velocity observations from Joughin et al.[38] and extract 1-dimensional time series of velocity at points spaced at 1 km intervals along a central flowline (as defined in ref. 44). We define an upstream limit to our analysis by the area for which there are sufficient velocity observations to constrain a time-continuous fit. The selected points are shown in Fig. 1A, B.

We then construct a continuous function that best fits the observed values at each point. Following Riel et al.[37], we perform a regularized least squares regression that estimates the optimal linear combination of representative time functions (linear polynomials, B-splines, and integrated B-splines of pre-defined center times and scales) to fit the data at each point. The resulting function is an optimized superposition of linear trend, seasonal variability, and secular change, which facilitates later decomposition into components of interest. For example, in the Results section labeled "Multi-annual..." we extract the long-term-varying signal to analyze cross-correlations of multi-annual change. Example observations and constructed continuous functions are shown in Fig. 1C.

### Normalized cross-correlation
Finally, we find and compare the cross-correlations describing ice speed response to each variable at each point. We sample each time-continuous function at regular intervals. Dickey-Fuller and KPSS tests applied using the Python package statsmodels v0.12.2[93] indicate that the raw time series are non-stationary − that is, their means and/or variances change over time, which can produce spurious results in cross-correlation analysis[94]. In sections "Seasonal to interannual..." - "No year in which...", we enforce stationarity by differencing:

$$f_i = \hat{f}_i - \hat{f}_{i-1}, \tag{1}$$

where $\hat{f}_i$ is the $i^{\text{th}}$ point in the raw time series and $f$ is the differenced time series. We elect not to difference the long-term-varying series tested in Results section "Multi-annual...", as doing so would remove the signal of interest.

We compute the normalized cross-correlation at lag $k$,

$$\text{XCorr}(f,v)_k = \frac{1}{N}\sum_{i=1}^{N}\frac{f_{i+k}-\bar{f}}{\sigma(f)}\frac{v_i-\bar{v}}{\sigma(v)}, \tag{2}$$

for $k \in [-N, N]$, where ice speed $v$ and variable $f$ are each time series of length $N$, differenced as in Eqn. (1), with means ($\bar{v},\bar{f}$) and standard deviations ($\sigma(v), \sigma(f)$). With this convention, a lag $k < 0$ refers to a cross-correlation with the velocity series offset backward in time; that is, strong cross-correlations at negative lag indicate that a change is observed first in the velocity signal and a similar change is observed later in the variable $f$ signal. The normalized cross-correlation may take values between ±1, and a cross-correlation at lag $k$ between two signals without autocorrelation is statistically significant at the 95% confidence level if it exceeds $1.96/\sqrt{N-k}$.

Each of the signals we study here includes some moderate to strong autocorrelation. Therefore, we correct the significance limits for each variable by a factor $\sqrt{(1+ab)/(1-ab)}$ where $a = 0.99$ is the lag-1 autocorrelation in the velocity signal and $b$ is the lag-1 autocorrelation for the variable $f$, following Dean and Dunsmuir[95].

Autocorrelation functions and resulting correction factors for each variable are shown in Fig. S2.

Because we anticipate multiple influences on observed surface velocity, we do not expect the magnitude of correlations to be close to 1. Rather, we identify the largest-magnitude statistically significant correlations for each variable at each point, and we compare their relative strength. From the full time series (Results, "Seasonal to interannual...") and then from annual subsets (Results, "No year...") and from series filtered to show only multi-annual variability (Results, "Multi-annual..."), we identify the largest magnitude of cross-correlation between the series and the lag in days at which that extreme value occurs. We restrict our analysis to positive lag values, consistent with determining which variables could be forcing (rather than responding to) velocity variability at Helheim (Supplementary Note S1). We present full correlograms with both positive and negative lag values in the supplement.

## Data availability

The terminus position data generated for this manuscript have been deposited in the Zenodo database, with access URL https://doi.org/10.5281/zenodo.5062050. The MEaSUREs velocity data used in this manuscript are publicly available through the National Snow and Ice Data Center: https://nsidc.org/data/NSIDC-0481/versions/1.

## Code availability

All code used in this analysis is available via GitHub and archived on Zenodo. Construction of the time-continuous velocity functions: https://doi.org/10.5281/zenodo.4474829. Along-flowline data extraction and cross-correlation: 10.5281/zenodo.4707999. Data pre-processing and visualization: https://doi.org/10.5281/zenodo.4707997.

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

## Acknowledgements
Work toward this manuscript was supported by the National Aeronautics and Space Administration grant NNX16AJ90G and Heising Simons Foundation grant 2017-316, both to L.S. The authors thank Brice P. Y. Noël for providing RACMO surface mass balance data and discussing its processing. L.U. thanks Jeremy Bassis for conversations about inference on weighted directed graphs, which informed the method used here. The authors also thank Kristin Poinar and Signe Hillerup Larsen, whose comments helped refine the manuscript.

## Author contributions
L.U. designed the study, with input from D.F. and B.M.; D.F. gathered model data and contributed literature review. L.S. contributed a dense record of satellite-derived terminus positions and guided its interpretation. B.R. developed software used to construct time-continuous velocity functions. L.U. performed quantitative analysis and produced manuscript figures. L.U. and D.F. drafted the manuscript, and all authors contributed to editing and approving its final form.

## Competing interests
The authors declare no competing interests.
