## [Peer Review File · Nature Communications]

Reviewer Comments, first round

Reviewer #1 (Remarks to the Author):

Reviewer report – Ultee et al., 2022

Key results:

This study, focussing on Helheim glacier, presents a statistical analysis of the relationship between the seasonal and inter-annual velocity variations at points along a flow-line, with three key variables: Surface Mass Balance, Runoff and Terminus position. The key findings are the different timescales of reaction, specific to Helheim glacier: runoff appears most important for seasonal velocity variations whereas terminus position is most important for inter-annual variability. Furthermore they find that the subglacial ridge, is an obstacle for travelling waves and that terminus position changes at Helheim in the period 2009-2016 are likely to be a response to rather than the cause of velocity changes.

Validity:

The study uses a simple cross-correlation between surface velocity and three key variables, which I see as a robust method to identify what variables should be key in projection modelling efforts such as ISMIP. And data treatment is described clearly.

Significance:

Raising the issue of timescales of reactions is timely in the context of recent studies of the effect, on seasonal to interannual timescales, of the surface runoff on marine terminating glaciers in Greenland as well as the recent ISMIP6 effort. The methodology brought forward can be applied to other glaciers based on the instructions in the paper, which gives good possibilities to test if the site specific conclusions for Helheim are also valid for other larger marine-terminating glaciers in Greenland in order to evaluate on the forcing suggested in ISMIP for example.

The site specific analysis for Helheim show insights into Helheim glacier, which is highly productive, yet maintains a relatively stable front position.

The finding that velocity changes drive calving front position changes at Helheim, shows how Helheim glacier front position is in a stable position which basically means that changes have yet not been large enough to bring the front position into an unstable bedrock geometry. And as also shown, the unusual stable position of Helheim glacier could be due to the fact that kinematic waves cannot travel very far inland.

Data and methodology

In my opinion the approach is robust.

Analytical approach

The conclusions made on the seasonal cross-correlation with runoff could benefit from some additional discussion, in my opinion. I am missing an explanation for the time lag of 60-90 days in the cross-correlation between velocity and runoff in the context of seasons.

Suggested improvements

Include discussion on the lag on the cross-correlation between runoff and velocity. Why is it 60 to 90 years, when studies at Helheim show instantaneous reactions to runoff in Andersen et al 2011 and 2011?

The integrated SMB over the large Helheim catchment is highly influenced by the large accumulation zone. If we disregard the high uncertainty on regional climate models representation of accumulation, the effect of the SMB might just be on timescales that are beyond seasonal or even decadal and thus beyond the investigated period here. This could perhaps be mentioned in the discussion?

Clarity and context

The text is clear and discussion thorough.

In line 110 there is referred to Text S3, but should that be S4?

References

I think the references are appropriate.

Your expertise

I am not an expert on statistical analysis and never tried making a cross-correlation analysis myself, and thus I am not able to evaluate whether a better approach exists. With my knowledge of the authors previous work, I am however, confident that they know what they are doing. This specific lack of personal expertise could be the reason for me needing an extra explanation on the time lag between runoff and velocity.

With best regards,
Signe Hillerup Larsen
shl@geus.dk

Reviewer #2 (Remarks to the Author):

Overview

This paper provides a quantitative analysis of drivers of ice velocity variability at Helheim Glacier. The novelty of this study lies its strictly quantitative approach, whereas controls on velocity previously often have been identified qualitatively, or aggregated for larger areas. Since Helheim Glacier is one of the highest-flux outlets of the Greenland Ice Sheet, identifying drivers of its velocity variation is highly relevant. Overall, the paper is very well written and the methodology and results have been provided with sufficient consideration of previous work. My main concerns are related to the relatively short study period for analyzing drivers of multi-annual flow variability, otherwise I only have minor comments.

Major comments

My main concern is whether the studied period 2009-2017 is sufficiently long period for studying the long-term varying velocity variability, since the velocities only have minor variations in this study period (Figure 1C). I get the impression that the authors agree that this is not very 'long-term' and have decided to only use the wording 'multi-annual' in the abstract and conclusions, but in the results section this is still called 'long-term'.

Furthermore, the last sentence in the abstract refers to negative lag correlations, which are only presented very short in Section 3.2 and Figure S3, but not in the Discussion or Conclusion. This

should be elaborated more in the main text if it is to be included in the abstract.

Only correlations with catchment-integrated SMB and runoff are studied, did you also consider integrating only the upstream part of the catchment for each point on the flowline?

There are figures with many panels, sometimes the reader would benefit from a bit more guidance to certain panels instead of the entire figure. Some suggestions are given in specific comments. Also, some figures contain a lot of information for different variables or locations, which makes them hard to read, especially if not zooming in to more than 100%. For example in Figure 3, it is hard to see in some panels whether there is a shading indicating significant correlations, or whether there are just many different correlations found for different locations.

Line by line comments

L45 Besides Sermeq Kujalleq, it would help to also refer to Jakobshavn Glacier/Isbrae in parenthesis.

L70 For 'scales', clarify whether referring to spatial and/or temporal scales.

L114 boxcar refers to a function, the resulting filter is called moving average.

L116 Add panels to reference, Figure 1E-G.

Figure 1 caption: stars in 1ABC are not explained and light curves in 1C are very faint. It seems like they are thinner than the light curves in E-G and thickening the lines in 1C could improve readability.

Figure 2 I find it a bit confusing that the same variable is once called Amax. Xcorr and (upper colorbar label) and once peak xcorr (lower colorbar label). I agree that it is visually appealing to show the peak xcorr and lag on a Hillshade map, but it might improve readability when plotting the values along an x-axis of distance along the flowline like in Figure 4A. This could be added as a supplementary Figure. In the current figure, I find it hard to say from the colorbar what lag the upper part of the peak xcorr for Catchment runoff has, which also seems close to 0.

L210 To me, the peak seems closer to a lag of 60 days than the 90 days mentioned in the text.

L221 Should mention in the text whether these negative lag correlations are significant, especially if this is to be included in the abstract.

L223-225 Maybe rephrase this sentence to explicitly state whether it is the positive-lag correlation of ice surface speed with terminus position which is weaker than the correlation with catchment-integrated runoff, or whether also the negative-lag correlation is weaker.

L231 Figures 2, 4A and S6

Figure 4A, is it really impossible to add the yticks in the figure instead of the caption? Also, it is a bit difficult to get an overview which correlations are shown in 4A and it might help to add a legend explaining what crosses, circles and diamonds stand for, instead of only describing all of this in the caption.

Section 3.4 does not mention that the sign change of absolute maximum cross-correlation is at a different location for terminus position than for SMB and runoff in Figure 2. I find it too simplified to state that the bedrock ridge as an obstacle for propagation of traveling waves explains this sign change for all three variables, especially since it is mentioned in Section 3.1 that a clear forcing of velocity by terminus position on the lower 10km of the glacier trunk is not apparent at the scale presented in this study. The difference between the 3 variables should be better explained in this section.

L244 Figure 4B

L282-297 the time periods studied by Moon et al. and Vijay et al should be mentioned in order to be able to compare the results.

Supplementary

Figure S1, what is the unit for lag here? Days or the ~ 3 day time difference between points in the time series?

Figure S6, right panels are slightly smaller due to colorbar.

Author Response to Reviews of

Helheim Glacier ice velocity variability responds to runoff and terminus position, but at different timescales

L. Ultee, D. Felikson, B. Minchew, L. A. Stearns, B. Riel
Nature Communications

RC: *Reviewer Comment*, **AR:** *Author Response*, Manuscript text

We thank the reviewers for their consideration. Both reviewers offered very constructive comments. We have implemented all suggested changes, specified below. We apologize for the delay in our response and appreciate the reviewers' patience.

1. Reviewer 1, Signe Hillerup Larsen

1.1. Key results

RC: *This study, focussing on Helheim glacier, presents a statistical analysis of the relationship between the seasonal and inter-annual velocity variations at points along a flow-line, with three key variables: Surface Mass Balance, Runoff and Terminus position. The key findings are the different timescales of reaction, specific to Helheim glacier: runoff appears most important for seasonal velocity variations whereas terminus position is most important for inter-annual variability. Furthermore they find that the subglacial ridge, is an obstacle for travelling waves and that terminus position changes at Helheim in the period 2009-2016 are likely to be a response to rather than the cause of velocity changes.*

AR: *We agree with the reviewer's summary of the work.*

1.2. Validity

RC: *The study uses a simple cross-correlation between surface velocity and three key variables, which I see as a robust method to identify what variables should be key in projection modelling efforts such as ISMIP. And data treatment is described clearly.*

AR: *Thank you.*

1.3. Significance

RC: *Raising the issue of timescales of reactions is timely in the context of recent studies of the effect, on seasonal to interannual timescales, of the surface runoff on marine terminating glaciers in Greenland as well as the recent ISMIP6 effort. The methodology brought forward can be applied to other glaciers based on the instructions in the paper, which gives good possibilities to test if the site specific conclusions for Helheim are also valid for other larger marine-terminating glaciers in Greenland in order to evaluate on the forcing suggested in ISMIP for example.*

The site specific analysis for Helheim show insights into Helheim glacier, which is highly productive, yet maintains a relatively stable front position.

The finding that velocity changes drive calving front position changes at Helheim, shows how Helheim glacier front position is in a stable position which basically means that changes have yet not been large enough to bring the front position into an unstable bedrock geometry. And as also shown, the unusual stable position of Helheim glacier could be due to the fact that kinematic waves cannot travel very far inland.

AR: *We thank the reviewer for this assessment.*

1.4. Data and methodology

RC: *In my opinion the approach is robust.*

AR: *Thank you.*

1.5. Analytical approach

RC: *The conclusions made on the seasonal cross-correlation with runoff could benefit from some additional discussion, in my opinion. I am missing an explanation for the time lag of 60-90 days in the cross-correlation between velocity and runoff in the context of seasons.*

AR: *We have added a paragraph describing possible physical interpretation of the lag time and explaining why we do not pursue it further:*

It is tempting to attribute physical meaning to the lag times of strongest cross-correlation. For example, the 60-day lag time in cross-correlation between runoff and ice surface speed that we observe in many years (Figure 3) could reflect the time required for water input to induce large-scale changes in subglacial water pressure, and therefore ice sliding velocity. That interpretation would agree with the model results of Poinar et al. (2019), who applied low-elevation meltwater input to an idealized Southeast Greenland outlet glacier and found that domain-averaged subglacial water pressure peaked 60 days into the melt season. The range of lag times for the handful of significant cross-correlations between speed and terminus position, 1-100 days, also aligns with theoretical results on the speed of dynamic wave propagation upstream from a calving terminus (Amundson et al., 2022). However, we caution that single-differencing the signals in Sections 3.1-3.2 produces different phase shifts in signals with different shapes (Supplementary Text S3 and Figure S4), which complicates the interpretation of lag times. We therefore refrain from more detailed interpretation of the lag times.

1.6. Suggested improvements

RC: *Include discussion on the lag on the cross-correlation between runoff and velocity. Why is it 60 to 90 [days], when studies at Helheim show instantaneous reactions to runoff in Andersen et al 2010 and 2011?*

AR: *We have added the above paragraph about interpreting lag times to the Discussion. The lag times in our analysis are not directly comparable to those in Andersen et al 2010 and 2011, since we use remote sensing data that has a more coarse temporal resolution than the field data they use. We have added a sentence to highlight this:*

Because we analyse remote sensing and climate model output data, the significant cross-correlations we find are at seasonal and longer timescales; this complements the shorter-timescale correlations previously found in field data (Nettles et al. 2008; de Juan et al. 2010; Andersen et al. 2010, 2011; Stevens et al. 2022).

We mention the distinction again later in the Discussion:

Field observations of Helheim Glacier at finer temporal scales than we study here have found that calving activity was an important control on velocity at the timescale of minutes to hours (Nettles et al., 2008; de Juan et al., 2010), and that runoff during the melt season contributes to daily velocity increases (Andersen et al. 2010, 2011; Stevens et al. 2022). Thus, even with our temporally dense records—average 3 days between measurements—we may more realistically expect to see responses to runoff than to iceberg calving. ... Extending our methodology to analyse the fine spatial and temporal scales captured in field observations could provide a fuller picture of the forcings driving velocity variability (building on Podrasky et al., 2012, for example).

RC: *The integrated SMB over the large Helheim catchment is highly influenced by the large accumulation zone. If we disregard the high uncertainty on regional climate models representation of accumulation, the effect of the SMB might just be on timescales that are beyond seasonal or even decadal and thus beyond the investigated period here. This could perhaps be mentioned in the discussion?*

AR: *Yes, this is a good point and it was echoed by the second reviewer’s comments about “long-term” versus “multi-annual” signals. We have added the following paragraph to the Discussion:*

The eight-year period of overlapping observations we studied, 2009-2017, necessarily limits our ability to resolve longer-term variability that may be important for glacier dynamics. For example, we found that surface mass balance had the lowest cross-correlations with velocity among the three variables we tested, but that finding does not preclude surface mass balance driving velocity variation at decadal and longer time scales. Ice core and radar reconstructions have shown that Greenland surface mass balance varies at multi-annual to multi-decadal time scales, correlated with the North Atlantic Oscillation, Atlantic Multidecadal Oscillation, and Greenland Blocking Index [1, 2]. The known long-term variability of surface mass balance, combined with Helheim Glacier’s large accumulation area and the long climatic response time of glacier flow [3, 4], suggest that correlations between surface mass balance and ice velocity may well be found in records longer than those we have studied here.

1.7. Clarity and context

RC: *The text is clear and discussion thorough.*

AR: *We appreciate the reviewer’s assessment.*

RC: *In line 110 there is referred to Text S3, but should that be S4?*

AR: *Yes, corrected. Thank you!*

1.8. References

RC: *I think the references are appropriate.*

AR: *We thank the reviewer for checking the references.*

1.9. Your expertise

RC: *I am not an expert on statistical analysis and never tried making a cross-correlation analysis my self, and thus I am not able to evaluate whether a better approach exists. With my knowledge of the authors previous work, I am however, confident that they know what they are doing. This specific lack of personal expertise could be the reason for me needing an extra explanation on the time lag between runoff and velocity.*

AR: *We appreciate the reviewer's consideration. We have added text to the Discussion, as shown above, to better communicate challenges in interpreting the lag. We hope this will be helpful for readers of similar background to the reviewer.*

2. Reviewer 2

2.1. Overview

RC: *This paper provides a quantitative analysis of drivers of ice velocity variability at Helheim Glacier. The novelty of this study lies its strictly quantitative approach, whereas controls on velocity previously often have been identified qualitatively, or aggregated for larger areas. Since Helheim Glacier is one of the highest-flux outlets of the Greenland Ice Sheet, identifying drivers of its velocity variation is highly relevant. Overall, the paper is very well written and the methodology and results have been provided with sufficient consideration of previous work. My main concerns are related to the relatively short study period for analyzing drivers of multi-annual flow variability, otherwise I only have minor comments.*

AR: *We thank the reviewer for this assessment, and for helpful suggestions below.*

2.2. Major comments

RC: *My main concern is whether the studied period 2009-2017 is sufficiently long period for studying the long-term varying velocity variability, since the velocities only have minor variations in this study period (Figure 1C). I get the impression that the authors agree that this is not very 'long-term' and have decided to only use the wording 'multi-annual' in the abstract and conclusions, but in the results section this is still called 'long-term'.*

AR: *We agree with the reviewer that 2009-2017 is too short to see variation over decadal and longer periods, and we have added a paragraph to the Discussion (L 415-428, quoted in response to Reviewer 1) to highlight that limitation—particularly important for variation that could be driven by surface mass balance, as another reviewer pointed out. In Results section 3.3, we use the term "long-term-varying" to distinguish the filtered velocity series from those with seasonal variation left unfiltered. We have edited the Methods section to make that connection more clear as well.*

RC: *Furthermore, the last sentence in the abstract refers to negative lag correlations, which are only presented very short in Section 3.2 and Figure S3, but not in the Discussion or Conclusion. This should be elaborated more in the main text if it is to be included in the abstract.*

AR: *We have added a short mention to the Discussion and referred readers to Supplementary Figure S3.*

In several years, there are significant cross-correlations between ice surface speed and terminus position, but they occur at negative lag times (Figure S3). That suggests that terminus position is responding to, rather than driving, seasonal velocity variability.

We have added a similar summary statement to the Conclusions:

The strongest cross-correlations between ice speed and terminus position occur at 0-day or negative lag times, suggesting that terminus position is responding to rather than driving seasonal velocity variability.

RC: *Only correlations with catchment-integrated SMB and runoff are studied, did you also consider integrating only the upstream part of the catchment for each point on the flowline?*

AR: Thank you for the suggestion. We considered it but determined that there would be substantially more data processing to find the unique catchment that drains to each point along the flowline, and that that data processing would address only one of several possible improvements to the data inputs. We have added text to the discussion (L399-414) to point out this limitation and others that may be addressed in future work:

Nevertheless, future studies could benefit from more sophisticated data processing or more detailed numerical modelling than we have presented here. For example, we produced time series of both surface mass balance and runoff integrated over the whole Helheim catchment. We suspect that integrating both fields over only the portion of the catchment that is upstream of each point along the flowline would provide more spatially refined information. The advantages to be gained by additional data processing, however, should be weighed against the major uncertainties remaining from unconstrained processes. For example, our work has not accounted for the changing state of the en/subglacial hydrologic system over the melt season, nor for the effect of the Helheim catchment's firn aquifer (Miller et al., 2020) on both glacial hydrology and dynamics. To do so would likely require the use of a glacial hydrology model (such as Werder et al., 2013). Future efforts could also explore the use of Bayesian methods to account for uncertainties in climate-model-derived runoff, which can be as large as 20% (van As et al., 2018; Noël et al., 2019).

RC: *There are figures with many panels, sometimes the reader would benefit from a bit more guidance to certain panels instead of the entire figure. Some suggestions are given in specific comments.*

AR: We have added reference to specific panels throughout. Thank you.

RC: *Also, some figures contain a lot of information for different variables or locations, which makes them hard to read, especially if not zooming in to more than 100%. For example in Figure 3, it is hard to see in some panels whether there is a shading indicating significant correlations, or whether there are just many different correlations found for different locations.*

AR: We have revised Figure 3 and S3 to better distinguish significance shading from the curves showing cross-correlations at different locations, by: (1) making the curves heavier, (2) making shading lighter, and (3) extending shading to cover the full range of the y-axis.

2.3. Line by line comments

RC: *L45 Besides Sermeq Kujalleq, it would help to also refer to Jakobshavn Glacier/Isbrae in parenthesis.*

AR: Added, thank you.

RC: *L70 For 'scales', clarify whether referring to spatial and/or temporal scales.*

AR: Added "time" scales.

RC: *L114 boxcar refers to a function, the resulting filter is called moving average.*

AR: Changed to "a 1-year moving average filter".

RC: *L116 Add panels to reference, Figure 1E-G.*

AR: Added, thank you.

RC: *Figure 1 caption: stars in 1ABC are not explained and light curves in 1C are very faint. It seems like they*

are thinner than the light curves in E-G and thickening the lines in 1C could improve readability.

AR: *We have revised the caption to clarify that the stars show locations for example velocity series plotted in 1C. We have also thickened the light curves in 1C.*

RC: ***Figure 2 I find it a bit confusing that the same variable is once called Amax. Xcorr and (upper colorbar label) and once peak xcorr (lower colorbar label). I agree that it is visually appealing to show the peak xcorr and lag on a Hillshade map, but it might improve readability when plotting the values along an x-axis of distance along the flowline like in Figure 4A. This could be added as a supplementary Figure. In the current figure, I find it hard to say from the colorbar what lag the upper part of the peak xcorr for Catchment runoff has, which also seems close to 0.***

AR: *We have corrected the colorbar labels so that both read “AMax. xcorr” - thank you for catching that. We have also corrected the colorbar label on Figure S6. Finally, we have added Figure S7, which shows the cross-correlations and lags together along the flowline. We direct readers to it in the caption of Figure 2. The lag at maximum cross-correlation between runoff and velocity in the upper part of the flowline is 70-80 days.*

RC: ***L210 To me, the peak seems closer to a lag of 60 days than the 90 days mentioned in the text.***

AR: *The reviewer is correct; we have confirmed that the lag time with runoff in 2009 is 63-66 days, depending on the point along the flowline. We have corrected the text accordingly.*

RC: ***L221 Should mention in the text whether these negative lag correlations are significant, especially if this is to be included in the abstract.***

AR: *We have revised to clarify that some of the negative-lag correlations are significant.*

RC: ***L223-225 Maybe rephrase this sentence to explicitly state whether it is the positive-lag correlation of ice surface speed with terminus position which is weaker than the correlation with catchment-integrated runoff, or whether also the negative-lag correlation is weaker.***

AR: *The negative-lag correlation is also weaker, as Figure S3 shows. We have added the following clarification:*

In most years and for most points, the correlation with runoff is stronger than that with terminus position for both positive and negative lags (Figure S3).

RC: ***L231 Figures 2, 4A and S6***

AR: *Changed, thank you.*

RC: ***Figure 4A, is it really impossible to add the yticks in the figure instead of the caption? Also, it is a bit difficult to get an overview which correlations are shown in 4A and it might help to add a legend explaining what crosses, circles and diamonds stand for, instead of only describing all of this in the caption.***

AR: *We have added tick labels and a legend in the figure, as the reviewer requested. We have also moved the variable names into the plot instead of along the y-axis in panel a, to reduce crowding.*

RC: ***Section 3.4 does not mention that the sign change of absolute maximum cross-correlation is at a different location for terminus position than for SMB and runoff in Figure 2.***

AR: *We do not see the difference in location of sign change that the reviewer mentions. We have not made corresponding changes to Section 3.4.*

RC: *I find it too simplified to state that the bedrock ridge as an obstacle for propagation of traveling waves explains this sign change for all three variables, especially since it is mentioned in Section 3.1 that a clear forcing of velocity by terminus position on the lower 10km of the glacier trunk is not apparent at the scale presented in this study. The difference between the 3 variables should be better explained in this section.*

AR: *We have added three new sentences to expand on expected dynamics of each of the three variables at the end of Section 3.4.*

We interpret that the bedrock ridge is an obstacle to the propagation of traveling waves (Nye, 1960; Fowler, 1982; Weertman and Birchfield, 1983). For example, adjustment in the glacier stress balance due to changes in ice accumulation (related to surface mass balance) would propagate as a kinematic wave from the accumulation zone to the ablation zone, and that wave could be obstructed by the vertical and lateral constriction of the ridge. Similarly, changes at the terminus can initiate upstream-propagating kinematic (Felikson et al., 2021) or dynamic waves (Amundson et al., 2022), which could be slowed by the steeper bed slopes around the ridge. We would also expect wave-like propagation of changing basal friction due to seasonal runoff input. The bedrock ridge modifies bed slope and ice overburden pressure, which will modify the hydraulic potential gradient and therefore also the direction of subglacial water flow around it.

RC: *L244 Figure*

AR: *Changed, thank you.*

RC: *L282-297 the time periods studied by Moon et al. and Vijay et al should be mentioned in order to be able to compare the results.*

AR: *We have added the time periods studied by these other papers to the text.*

RC: *Figure S1, what is the unit for lag here? Days or the 3 day time difference between points in the time series?*

AR: *It was the latter. We have updated the figure to show lag in days, consistent with other figures.*

RC: *Figure S6, right panels are slightly smaller due to colorbar.*

AR: *Fixed! Thank you for pointing this out. We have also corrected the colorbar labels and caption, for consistency with main text Figure 2.*

References

- [1] Mernild, S.H., Hanna, E., McConnell, J.R., Sigl, M., Beckerman, A.P., Yde, J.C., Cappelen, J., Malmros, J.K. and Steffen, K. (2015), Greenland precipitation trends in a long-term instrumental climate context (1890–2012): evaluation of coastal and ice core records. *Int. J. Climatol.*, 35: 303-320. <https://doi.org/10.1002/joc.3986>
- [2] Lewis, G., Osterberg, E., Hawley, R., Whitmore, B., Marshall, H. P., and Box, J. (2017). Regional Greenland accumulation variability from Operation IceBridge airborne accumulation radar, *The Cryosphere*, 11, 773–788. <https://doi.org/10.5194/tc-11-773-2017>

- [3] Nye, John Frederick (1960). The response of glaciers and ice-sheets to seasonal and climatic changes. *Proc. R. Soc. Lond. A* 256, 559–584. <http://doi.org/10.1098/rspa.1960.0127>
- [4] Harrison, W., Elsberg, D., Echelmeyer, K., Krimmel, R. (2001). On the characterization of glacier response by a single time-scale. *Journal of Glaciology*, 47(159), 659-664. doi:10.3189/172756501781831837

Reviewer Comments, second round

Reviewer #2 (Remarks to the Author):

I would like to thank the authors for their thorough and clear response to my feedback. I have no further comments to this work, which I find to be an important contribution to our understanding of tidewater glacier dynamics.

Author Response to Reviews of

Helheim Glacier ice velocity variability responds to runoff and terminus position change at different timescales

L. Ultee, D. Felikson, B. Minchew, L. A. Stearns, B. Riel
Nature Communications

RC: *Reviewer Comment*, **AR:** *Author Response*, Manuscript text

Only one reviewer made any comment. For completeness, we respond directly below.

1. Reviewer 1, Signe Hillerup Larsen

AR: *Reviewer 1 did not comment during this round. We thank the reviewer for her earlier constructive comments.*

2. Reviewer 2

RC: *I would like to thank the authors for their thorough and clear response to my feedback. I have no further comments to this work, which I find to be an important contribution to our understanding of tidewater glacier dynamics.*

AR: *We appreciate the reviewer's positive assessment. We thank the reviewer for their service.*